# Calcium Signaling-Mediated Plant Response to Cold Stress

**DOI:** 10.3390/ijms19123896

**Published:** 2018-12-05

**Authors:** Peiguo Yuan, Tianbao Yang, B.W. Poovaiah

**Affiliations:** 1Laboratory of Molecular Plant Science, Department of Horticulture, Washington State University, Pullman, WA 99164-6414, USA; pomology2010@gmail.com; 2United States Department of Agriculture (USDA)-Agricultural Research Service (ARS), Food Quality Laboratory, Beltsville, MD 20705, USA; Tianbao.Yang@ARS.USDA.GOV

**Keywords:** plants, calcium signaling, cold stress response

## Abstract

Low temperatures have adverse impacts on plant growth, developmental processes, crop productivity and food quality. It is becoming clear that Ca^2+^ signaling plays a crucial role in conferring cold tolerance in plants. However, the role of Ca^2+^ involved in cold stress response needs to be further elucidated. Recent studies have shown how the perception of cold signals regulate Ca^2+^ channels to induce Ca^2+^ transients. In addition, studies have shown how Ca^2+^ signaling and its cross-talk with nitric oxide (NO), reactive oxygen species (ROS) and mitogen-activated protein kinases (MAPKs) signaling pathways ultimately lead to establishing cold tolerance in plants. Ca^2+^ signaling also plays a key role through Ca^2+^/calmodulin-mediated *Arabidopsis* signal responsive 1 (AtSR1/CAMTA3) when temperatures drop rapidly. This review highlights the current status in Ca^2+^ signaling-mediated cold tolerance in plants.

## 1. Introduction

Cold is a major environmental factor that limits plant growth and reduces productivity and quality [1]. Under low temperature conditions, plants exhibit a variety of cold-induced physiological and biochemical responses, such as production of reactive oxygen species (ROS), changes in membrane lipid composition and changes in osmolytes [1,2]. Low temperatures also induce the expression of certain *cold regulated* (*COR*) genes, such as *responsive to desiccation 29* (*RD29A*), COR15a, and *kinase* (*KIN1*), in order to stabilize membranes against freezing-induced injury [1]. The dehydration response element-binding protein (DREB1) or CRT/DRE-binding factors are a family of closely related AP2/ERF transcription factors that regulate the *COR* gene expression [3], and *c-repeat binding factors*
*(CBFs)* are themselves upregulated by cold through CBF expression 1 (ICE1) [4,5].

Plant response to cold stress is a complex process. This response has been reported to require several signal pathways such as oxidative pathway, mitogen-activated protein kinases (MAPKs), phytohormone, abscisic acid (ABA), as well as *Arabidopsis* response regulators (ARRs) [6,7,8,9]. Ca^2+^, as a second messenger, is known to be involved in a variety of biological processes in eukaryotic cells, including playing a critical role in cold stress response in plants [8,10,11]. Plants perceive cold stimulus by sensing the changes in cytoplasmic membrane via cold signal sensors, such as chilling tolerance divergence 1 (COLD1) [12]. Cold signal transduction involves the activation of Ca^2+^ channels and/or Ca^2+^ pumps to induce Ca^2+^ influx (Ca^2+^ signature) in plant cells [13,14]. Ca^2+^ signals triggered by cold stimulus are relayed by Ca^2+^ sensors, such as calmodulins (CaMs), CaM-like proteins (CMLs), Ca^2+^-dependent protein kinases (CPKs/CDPKs), and calcineurin B-like proteins (CBLs) [8,14,15,16]. These Ca^2+^ sensors, together with other components in Ca^2+^ signaling, decode Ca^2+^ signals into downstream signaling events, such as phosphorylation, transcriptional reprogramming, activation of MAPKs cascade, as well as the accumulation of ROS or nitric oxide (NO) [17,18], suggesting that Ca^2+^ signaling plays a key role in mediating plant response to cold stress.

## 2. Cold Stress-Induced Calcium Transients

Changes in Ca^2+^ transients are an early event in plant response to diverse environmental signals. The specific characteristics of the Ca^2+^ transients are defined as Ca^2+^ signatures, including amplitude, duration, and frequency [19]. It is well documented that microbial signals induce specific Ca^2+^ signatures in the root hair [20,21]. For example, the fungal symbiotic microbe (mycorrhizae) triggers a different Ca^2+^ signature, as compared to bacterial symbiotic microbe (rhizobacteria) [10,22,23,24]. Similarly, low temperatures also trigger Ca^2+^ transients in plants [25,26]. Aequorin-based Ca^2+^ imaging revealed that low temperature (0 °C) induced a rapid and transient Ca^2+^ influx in whole seedlings of *Arabidopsis thaliana* [27]. Transient Ca^2+^ changes triggered by cold stress were also observed by using another Ca^2+^ sensor; Yellow Cameleon (YC3.6) [28]. The patch-clamp technique provided evidence of a predicted nonselective Ca^2+^-permeable cation channel in *Arabidopsis* mesophyll cells, and this Ca^2+^ channel is activated by cold stress [25].

## 3. Membrane Proteins Involved in the Perception of Cold Stress Signal Trigger Ca^2+^ Transients

Perception of environmental stimuli is considered an upstream event involved in inducing Ca^2+^ transients. Plants use diverse cytoplasmic membrane-localized receptor proteins/kinases in the perception and transduction of developmental and environmental signals. Although numerous receptor proteins/kinases (RKs) or receptor-like proteins/kinases (RLKs) have been identified in the perception of specific pathogens or hormones, few RKs or RLKs have been observed in the perception of cold stress signals in plants. Previous studies have indicated that plants could perceive cold signals through the physical change or damage of the cellular membranes [29,30]. For example, low temperatures (0–10 °C) cause the reduction of fluidity in the cellular phospholipid membrane, while chilling stress results in plasma membrane rigidity [31]. Supporting evidence is that dimethyl sulfoxide (DMSO) triggers plant cell membrane rigidification and activates a cold response signaling pathway even when plants are grown at normal temperatures, while benzyl alcohol, as a fluidizer of cell membrane, suppresses a cold response signaling pathway when plants are grown at low temperatures [8]. It is a reasonable assumption that the damage to plant membranes leads to ion leakage that induces Ca^2+^ changes under low temperature conditions. However, the perception, transduction, and the ultimate response to cold signals are complex and not clearly understood.

Another assumption is that the accumulation of ROS triggered by cold stress results in Ca^2+^ transients in plant cells. Low temperatures cause the production of ROS, and enhanced ROS induces Ca^2+^ transients, although the mechanisms are not clear [32,33]. Previous reports identify that respiratory burst oxidase homologs (RBOHs) and encoding plasma membrane NADPH oxidases are crucial for ROS burst in plant response to diverse environmental stresses, including cold stress [34,35,36]. Further study suggests that soybean genes regulated by cold 2 (AtSRC2) interacts with N-terminal of AtRBOHF and activates Ca^2+^-mediated NADPH oxidase activity of AtRBOHF in response to cold stress [37]. These observations suggest that the cross-talk between Ca^2+^ signaling and ROS signaling plays a role in the perception of cold stimulus.

A recent study identified a specific receptor, COLD1, involved in the recognition of cold stress signal in rice. COLD1 is a plasma membrane (PM)- and endoplasmic reticulum (ER)-localized transmembrane protein which senses cold signals and induces cold stress tolerance in plants [38]. The *cold1* knock-out mutants display cold sensitivity as compared to WT, when grown at 4 °C for 96 h; while the *cold1-*overexpressed plants display cold tolerance under similar conditions. In addition, Ca^2+^ transients triggered by cold stress are compromised in *cold1* mutants as compared to WT. However, the hierarchy of Ca^2+^ transient changes and COLD1 in plant cold stress signaling pathway is unclear [12,39]. One reasonable explanation is that *COLD1* encodes a cation permeable channel or a subunit of calcium channel.

In addition, a recent study revealed that two putative Ca^2+^-permeable mechanosensitive channels, mid1-complementing activity 1 (MCA1) and MCA2, are involved in cold-induced Ca^2+^ transients in *Arabidopsis* [13]. MCA1 and MCA2 are PM-localized cation channels, and MCA1 interacts with MCA2 to form a homotetramer to regulate Ca^2+^ increase in the cytoplasm [40,41,42]. Ca^2+^ influx triggered by cold stress is compromised in *mca1 mca2* double mutant plants. Also, this double mutant plant displays cold sensitivity as compared to wild-type (WT). Further study revealed that MCA1 and MCA2 regulated CBF-independent cold signaling in plants, although the mechanism remains unclear [13]. 

## 4. Ca^2+^ Signaling-Mediated Phosphorylation Events Involved in Cold Stress Response

### 4.1. Ca^2+^/CaM-Regulated Receptor-Like Kinases 

It has been reported that Ca^2+^/CaM-regulated receptor-like kinases 1 (CRLK1), encoding a PM-associated serine/threonine kinase regulated by Ca^2+^ signaling, plays a critical role in plant response to cold stress [43,44,45,46]. *crlk1* knock-out mutant plants display enhanced sensitivity as compared to WT at freezing temperatures. The induction of cold-responsive genes triggered by cold stress, such as *CBF1*, *RD29A* and *COR15a*, were suppressed in *crlk1* plants as compared to WT. In addition, low temperature (4 °C) and 10 mM H_2_O_2_ treatments induced the expression of CRLK1 protein. These observations suggest that CRLK1 positively regulates cold response in *Arabidopsis*. Further studies suggest that Ca^2+^/CaM is required for the activation of CRLK1 kinase. The CRLK1 kinase activity increased with an increase in CaM concentration in the presence of Ca^2+^, while CaM antagonist, CPZ, suppressed the CaM-stimulated CRLK1 kinase activity. Furthermore, the CaM-binding domain in the C-terminal of CRLK1 (residues 369–390) is required for CaM-stimulated kinase activity, and only the specific CaM isoform, such as potato calmodulin 1 (PCM1) and not PCM6, is responsible for the activation of CRLK1. These observations suggest that there is a Ca^2+^ signaling-mediated cold response pathway regulated by CRLK1 [43,44].

### 4.2. Ca^2+^/CaM-Mediated CRLK and MAPK Cascade Response to Cold Stress

MAPK cascades are evolutionarily conserved in eukaryotic organisms [9,47,48,49]. Previous studies have suggested that MAPK signaling pathways play a positive role in the regulation of cold response in plants, since *mkk2* mutant plants are more sensitive to cold stress as compared to WT, while *mkk2*-overexpressed mutant plants display enhanced cold tolerance [45,46]. Additionally, the transcriptional expression of *CBF* was induced in *mkk2*-overexpressing mutant plants even at normal temperatures. Further study revealed that low temperature activates Ca^2+^ signal-mediated CRLK1, which subsequently phosphorylated and activated MEKK1, an upstream component in MAPK cascade of MKK2 [43,44,45,46]. Activation of MKK2 by MEKK1 induced phosphorylation of MPK4 and MPK6. These results suggest that Ca^2+^ signaling plays a key role in MAPK-mediated plant response to cold stress. A recent study revealed that MPK3 and MPK6 negatively regulate cold tolerance in plants through the phosphorylation of the inducer of CBF expression 1 (ICE1), which subsequently induces ICE1 degradation [50]. However, CRLK1 and CRLK2 repress the activation of MPK3 and MPK6 during cold stress, although the molecular mechanism is not clear. One reasonable assumption is that CRLKs regulates MEKK1-MKK1/2-MPK4 cascade (MEKK1 is reported to be phosphorylated by CRLK1 [44]), while MEKK1-MKK1/2-MPK4 cascade negatively regulates MPK4/MPK6 cascade [51]. In addition, as mentioned above, Ca^2+^ signaling also positively regulates MAPKs-mediated plant response to cold stress through CPKs.

### 4.3. Ca^2+^-Dependent Protein Kinases 

CPKs are reported to be involved in Ca^2+^-mediated signal transductions by connecting cold stress-triggered Ca^2+^ transients to downstream phosphorylation events [52,53,54,55]. In rice, OsCPK27 was identified as a positive regulator of cold tolerance. The transcriptional expression of OsCPK27 was greatly induced by cold stress (4 °C, 12 h) and *oscpk27*-silenced mutant plants displayed compromised cold tolerance as compared to WT [56]. Further study revealed that OsCPK27 induces cold response involving ROS, NO, and MAPKs pathways. At low temperatures, accumulations of ROS and NO was attenuated in *oscpk27*-silenced mutant plants. In addition, the activation of MAPK cascades, such as MPK1/2, was repressed in *oscpk27*-silenced mutant plants [56]. Another study revealed that OsCPK24 regulated low temperature-triggered production of ROS through direct phosphorylation and suppression of thioltransferase activity of glutaredoxin 10 (OsGrx10), which acts as a glutathione-dependent thioltransferase in ROS signaling pathways [57]. In addition, OsCPK17 was involved in cold stress and induced the accumulation of NO through its putative substrate, nitrate reductase 1 (OsNR1), involving NO metabolism [52]. OsCPK17 has been reported to confer cold tolerance through the activation of PM-localized water transport plasma membrane intrinsic protein 2-1 (OsPIP2;1) [58]. 

### 4.4. CBL and CIPK

CBLs, as Ca^2+^ signal sensors, also relay cold stress-triggered Ca^2+^ transients into phosphorylation events through their interaction with CBL-interacting protein kinases (CIPKs) [59]. CBLs and CIPKs play crucial roles in abiotic and biotic stresses [59,60,61]. In *Arabidopsis*, the transcriptional expressions of *CBL1* and *CIPK7* were induced at 4 °C, 3 h and 12 h, respectively [62]. In addition, *cbl1* knock-out mutant plants displayed increased cold sensitivity. In addition, the inductions of cold response genes, such as *COR15a* and *KIN1*, were suppressed in *cbl1*. Furthermore, CBL1 is found to interact with CIPK7 in the presence of Ca^2+^ in vitro and in vivo. These observations indicate that CIPK7, together with CBL1, are required for cold tolerance. In rice, the transcriptional expression of *OsCIPK1*, *OsCIPK3* and *OsCIPK9* were induced at low temperatures. Further study revealed that *oscipk3*-overexpressing mutant plants are more tolerant to cold stress as compared to WT, indicating that OsCIPK3 would be a positive regulator in cold tolerance [63]. A recent investigation of the entire signal networks of CBLs-CIPKs in response to cold stress in *Cassava*. *MeCBL2*, *MeCBL4*, *MeCBL5*, *MeCBL9*, and *MeCBL10* were induced by cold stress in roots, while only *MeCBL5* was induced by cold stress in leaves. In addition, *MeCIPK7*, *MeCIPK10*, and *MeCIPK13* were induced by cold stress in roots, while *MeCIPK4*, *MeCIPK12*, and *MeCIPK14* were induced by cold stress in leaves [64]. These observations suggest tissue-specific functions of CBLs-CIPKs in cold tolerance. However, how CIPKs and their substrates are involved in plant response to cold stress is not clear.

## 5. AtSR1/CAMTA3-Mediated Transcriptional Reprogramming

Ca^2+^/CaM-mediated transcription factors relay Ca^2+^ transients triggered by cold stress to transcriptional reprogramming. *Arabidopsis* signal responsive 1 (AtSR1), also known as calmodulin-binding transcriptional activator 3 (CAMTA3), is well documented as a CaM-mediated transcription factor (TF) to regulate gene expressions by binding to the “CGCG” DNA-binding motif in their promoter region [65,66,67]. AtSR1 positively regulates plant cold tolerance, since *atsr1* knock-out mutant plants display increased sensitivity to cold stress as compared to WT. The transcriptional expression of *CBF1* was suppressed in *atsr1* at low temperatures, and the promoter of *CBF1* contains the AtSR1-recognized DNA motif [68]. These observations indicate that AtSR1 regulates CBF1-mediated signaling pathway during cold stress. Furthermore, accumulated salicylic acid (SA) in *atsr1* had no impact on cold tolerance, but SA plays a key role in transcriptional reprogramming at low temperatures [69,70]. Promoter analysis of wound-induced genes revealed a rapid stress response DNA element (RSRE), CGCGTT. In addition, promoter activity assay revealed that luciferase activity level triggered by cold stress was compromised in *atsr1* as compared to WT [71], indicating AtSR1-mediate cold tolerance through the regulation of genes containing RSRE in their promoters. An earlier study indicated that AtSR1 positively regulates the transcriptional expression of CBF2 through binding to “CCGCGT” motif in its promoter region [68]. In addition, heptahelical protein 2 (HHP2) was identified to interact with AtSR1; HHP2, including HHP1 and HHP3, was induced by R2R3-type MYB transcription factor, MYB96, at low temperatures [72]. These results suggest that Ca^2+^ signaling regulates transcriptome response to cold stress through AtSR1.

A recent study revealed that AtSR1, together with CAMTA5, regulated CBFs only in response to a rapid reduction in temperatures, but not to a gradual reduction in temperatures (3 °C/10 min) [73]. Interestingly, unlike circadian clock associated 1 (CCA1), which only mediates cold response during the daytime, AtSR1 regulates cold response during both day and night-time [73], although the underlying mechanism is not clear. Another recent study revealed that the first IQ motifs in AtSR1 (residues 850–875) are required for its activity [74]. Additionally, post-translational modifications, such as phosphorylation or dephosphorylation, would play an important role in AtSR1-mediated signaling response to environmental stimulus. Two putative phosphorylation sites in AtSR1, S454, and S964, were identified [75]. Functional tests revealed that *camta1 camta3* double mutants complemented with mutated AtSR1 protein, S454A and S964A (in the phosphorylation sites of AtSR1), were partially restored to WT; and the repression of the biosynthesis of SA in the mutants was impaired, indicating that phosphorylation is required for the full function of AtSR1 [74]. 

It has been documented that AtSR1 is a defense repressor and is degraded to induce SA-mediated immune response during pathogen attack [67,76,77,78]. Interestingly, SA-mediated signaling pathways were also observed to be activated in plants subjected to long-term cold treatments (4 °C, one or two weeks) [79,80]. However, the activation of the SA signaling pathway was not caused by the degradation of AtSR1 protein. In contrast, AtSR1 protein accumulates under low temperatures [74]. These observations indicate that plants may possess multiple pathways to overcome AtSR1 suppression of the SA signaling pathway, in order to establish appropriate response to specific stress, although the molecular mechanism remains unclear.

## 6. Concluding Remarks and Future Directions

In recent years, the mechanism of plant cold stress tolerance and genes involved in the cold stress signaling network have been intensively investigated. Several signal pathways are involved in cold stress signaling. Perception of cold stimulus is considered to be the earliest event in triggering the induction of Ca^2+^ transients in plant cells [12]. A variety of Ca^2+^ channels and/or Ca^2+^ pumps are involved in creating cold stress-triggered Ca^2+^ transients, which are relayed and deciphered by various Ca^2+^ sensors to induce the changes in gene expression and eventually cold tolerance in plants [12,13]. Much progress has been made in the understanding of several Ca^2+^-mediated signal networks, such as Ca^2+^-CBL-CIPK and Ca^2+^-CaM-AtSR1/CAMTA3 [62,73,74]. Figure 1 illustrates the current status of Ca^2+^-mediated signaling in cold stress response. It should be noted that we attempted to cover as many studies on calcium-regulated cold signaling as possible. However, in this short communication we had to restrict ourselves to some recent highlights. Plant cold tolerance is a complex process involving a variety of signal transduction pathways. It is still not clear how Ca^2+^-mediated signaling interacts with other signaling pathways and coordinates cold response in plants. Furthermore, it is a challenge to dissect the role of Ca^2+^ signals in cold acclimation, and to determine whether cold stress induces Ca^2+^ transients in the nucleus. Emerging technologies, such as more sensitive Ca^2+^ imaging, -omics and gene editing approaches will be powerful tools to answer these questions. 

## Figures and Tables

**Figure 1 ijms-19-03896-f001:**
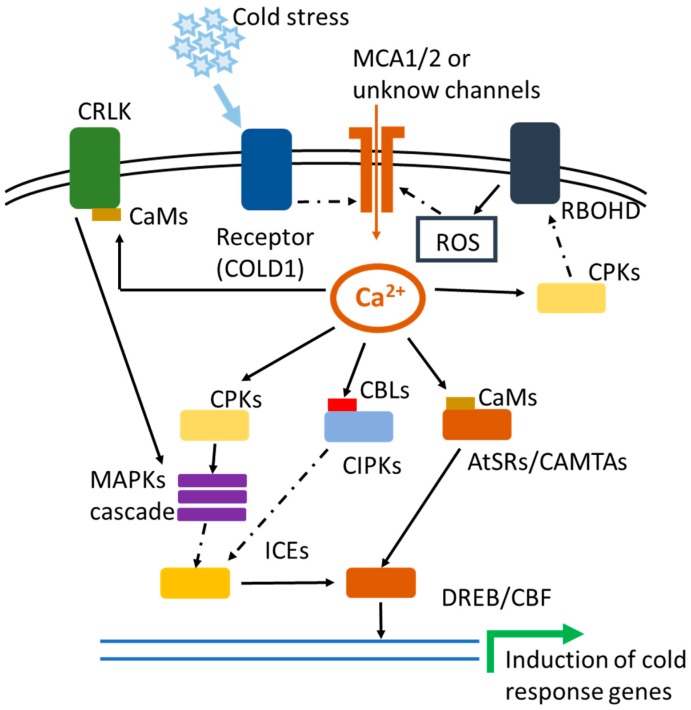
Calcium signaling-mediated plant response to cold stress in *Arabidopsis*. The perception of cold stimulus is the first step in the activation of cold tolerance in plants. Plants sense cold signals through the recognition of the changes in cellular membrane by cold stress, or cold signals sensor, COLD1. The perception of cold stimulus activates cold responsive Ca^2+^ channels (MCA1 and MCA2) or other unknown Ca^2+^ channels and/or pumps to induce Ca^2+^ transients, also knowns as Ca^2+^ signals or signatures. The cold stress-induced Ca^2+^ transient changes in plant cell and the expression of *AtSRC2* subsequently facilitate the production of ROS, through the activation of Ca^2+^-mediated NADPH oxidase activity of AtRBOHF. Enhanced ROS further activates Ca^2+^ channels and/or pumps for inducing Ca^2+^ transients to form a positive feedback. CPKs also relay cold-triggered Ca^2+^ signals into phosphorylation and activation of the MAPK cascade. In addition, the MAPK cascade is activated by CRLK1 or CRLK2 through the interaction with calmodulin (CaM) in response to cold stress. The MAPK signaling pathway triggered by cold stress suppresses the degradation of ICE1 to establish cold tolerance in plants. CaM also relays cold stress-triggered changes into transcriptional reprogramming via AtSR1/CAMTA3. Solid arrows represent physical interaction and/or positive regulation (induction). Dashed arrows represent mechanism that is unclear.

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
