# Peer review of "Calcium Signaling-Mediated Plant Response to Cold Stress"

_ijms, 2018, doi:10.3390/ijms19123896_

Reviewer 1 Report

The review of Peiguo et al. dials with the plant response to low temperature, focusing in particular on the role of intracellular calcium and on the calcium signalling pathway. The topic is interesting however, in the present version looks like a series of points with not so much logic among them. This referee found this paper difficult to read.

Major points:

- the paragraphs should be re-written following the relative title.

In paragraph 2 (line 40) the title is cold-stress-induce calcium transients. However, instead to give information about the duration, shape, modulation of the calcium transients, the authors discussed about calcium and mechanosensitive channels, which should be pleced in paragraph 3.

In paragraph 4 the order is confusing, 4.1 and 4.4 should be merged or at least be near;

- the bibliography should be carefully checked:

line 32: [10] looks not related to the discussed point, as well as, in line 72 [31] and in line 124 [50];

line 48 and 49 change of fluidity? where is it written in [21]?

- all the abbreviations should be defined and some words should be used to explain the physiological role.

Examples:

- line 24 COR

- line 25 DREB

- paragraph 4.4, mkk1

- line 164 TF

- line 172 LUC activity

- in the figure the ROS production should be included.

- all the text should be carefully revised

(i.e.,line 202, cold acclamation)

Author Response

Reviewer 1:

Major points:

In paragraph 2 (line 40) the title is cold-stress-induce calcium transients. However, instead to give information about the duration, shape, modulation of the calcium transients, the authors discussed about calcium and mechanosensitive channels, which should be pleced in paragraph 3.

Response: Thanks to the reviewer for these suggestions. We discussed calcium signature traits and placed the mechanosensitive channels in paragraph 3.

In paragraph 4 the order is confusing, 4.1 and 4.4 should be merged or at least be near;

Response: We have moved 4.4 after 4.1 and renamed as 4.2

the bibliography should be carefully checked:

line 32: [10] looks not related to the discussed point, as well as, in line 72 [31] and in line 124 [50];line 48 and 49 change of fluidity? where is it written in [21]?

Response: We revised these lines and removed line 72 [31]. Also, we agree with the reviewer that the previous version in lines 48 and 49 was not easy to understand and we revised it to make it more fluid. 

all the abbreviations should be defined and some words should be used to explain the physiological role.

Examples:

- line 24 COR

- line 25 DREB

- paragraph 4.4, mkk1

- line 164 TF

- line 172 LUC activity

Response: We have revised them accordingly and put the abbreviation list in the manuscript.

 in the figure the ROS production should be included.

Response: We have added ROS in fig 1.

all the text should be carefully revised (i.e.,line 202, cold acclamation)

Response: We have carefully revised all the text.

Reviewer 2 Report

 The manuscript is well written and organized. However, there are several comments for MDPI publication.

Does the phosphorylation and CAMTA3- mediated signaling have covered calcium -mediated cold response of plants? What about adding brief explanations in terms of phosphorylation in the introduction and abstracts.

In the introduction, author have explained an example. Line 42 to line 44, though this example is an available, this is too specific and it seems not to be linked with cold response. In addition, check the reference for this.  

In the introduction, authors mentioned that MCA1 and CBF2 regulates a CBF-independent cold signaling. However, there is not any contents about CBF independent signaling in the manuscript.

Connected with No. 3, CBFs are regulated several other factors. ICE is not a unique and all about it. Please see HHP and MYP96, if authors will describe the regulation of CBFs expression

Please add references at the sentences, line 99 and 100.

Author Response

Reviewer 2:

Does the phosphorylation and CAMTA3- mediated signaling have covered calcium -mediated cold response of plants? What about adding brief explanations in terms of phosphorylation in the introduction and abstracts.

Response: There are a lot of calcium-binding proteins in plants with a variety of functions.  In the review, we only highlighted the best characterized calcium-regulated proteins involved in cold tolerance. Note that even though we tried to cover all the calcium-regulated cold signaling reports, we could not include all of them. We added this sentence in the conclusion section.

We are not sure what the review wants for an explanation on phosphorylation. We discussed phosphorylation events as appropriate.

In the introduction, author have explained an example. Line 42 to line 44, though this example is an available, this is too specific and it seems not to be linked with cold response. In addition, check the reference for this. 

Response: We discussed fungi- and bacteria-triggered calcium changes since they are known to be the best characterized calcium signatures/changes.  We have double-checked the references.

In the introduction, authors mentioned that MCA1 and CBF2 regulates a CBF-independent cold signaling. However, there is not any contents about CBF independent signaling in the manuscript.

Response: In this mini-review, we highlighted calcium-regulated cold response. Thus, we have not discussed the details that are not closely involved in calcium signaling and cold response.

Connected with No. 3, CBFs are regulated several other factors. ICE is not a unique and all about it. Please see HHP and MYP96, if authors will describe the regulation of CBFs expression

Response: We did not discuss these because there is no strong evidence showing calcium signal-mediated HHP and MYP96.

Please add references at the sentences, line 99 and 100.

Response: We have added these references.

Reviewer 3 Report

The communication by Peiguo et al. is an interesting and well-written summary of the Ca2+-mediated response to cold stimuli.

I have only minor issues with this manuscript:

1) Authors state that there are several cold stress signalling pathways (concluding remarks), but the reference is missing. I believe that the overview of known cold-signalling networks should be provided in the introduction part, including links to phytohormonal networks (e.g. cold-responsive ARRs).  

2) Please, check the provided references. For example, the reference to DMSO-triggered membrane rigidification [7] is incorrect (most likely Orvar et al. 2001 should be referenced there). 

3) line 39 - I believe that "response to cold stress" is more appropriate than "cold tolerance". 

4) The list of abbreviations is not included.

5) The model presented in the figure is based on Arabidopsis, include this into its description.

Author Response

Reviewer 3:

1) Authors state that there are several cold stress signalling pathways (concluding remarks), but the reference is missing. I believe that the overview of known cold-signalling networks should be provided in the introduction part, including links to phytohormonal networks (e.g. cold-responsive ARRs). 

Response: We have added this and included references of other signal pathways in the introduction section.

2) Please, check the provided references. For example, the reference to DMSO-triggered membrane rigidification [7] is incorrect (most likely Orvar et al. 2001 should be referenced there).

Response: We have corrected this.

3) line 39 - I believe that "response to cold stress" is more appropriate than "cold tolerance".

Response: We have revised this sentence.

4) The list of abbreviations is not included.

Response: We have added the list of abbreviations.

5) The model presented in the figure is based on Arabidopsis, include this into its description.

Response: We have revised this.

Round  2

Reviewer 1 Report

After corrections, the quality of the paper improved. 

Author Response

Reviewer 2:

After corrections, the quality of the paper improved.

Response: We appreciate the reviewer’s positive comment.